# XLS-R fine-tuning on noisy word boundaries for unsupervised speech segmentation into words

**Robin Algayres**[1,2]**, Pablo Diego-Simon**[1]**, Benoit Sagot**[2]**, Emmanuel Dupoux**[1,3]

[1]ENS, INSERM, UPEC, PSL [2]Inria, Paris, [3]Meta AI

`robinalgayres@inria.fr`

## Abstract

Due to the absence of explicit word boundaries in the speech stream, the task of segmenting spoken sentences into word units without text supervision is particularly challenging. In this work, we leverage the most recent self-supervised speech models that have proved to quickly adapt to new tasks through fine-tuning, even in low resource conditions. Taking inspiration from semi-supervised learning, we fine-tune an XLS-R model to predict word boundaries themselves produced by top-tier speech segmentation systems: DPDP, VG-HuBERT, GradSeg and DP-Parse. Once XLS-R is fine-tuned, it is used to infer new word boundary labels that are used in turn for another fine-tuning step. Our method consistently improves the performance of each system and sets a new state-of-the-art that is, on average 130% higher than the previous one as measured by the F1 score on correctly discovered word tokens on five corpora featuring different languages. Finally, our system can segment speech from languages unseen during fine-tuning in a zero-shot fashion[1].

## Introduction

In an attempt to model infant ability to segment speech into words, researchers have aimed at unveiling word boundaries directly from the speech signal without prior knowledge of the language and, of course, without relying on textual annotations. Even though the notion of 'word' does not obey a set of strict rules, the goal is to bridge the existing large performance gap between speech-based and text-based segmentation systems (Dunbar et al., 2022)[2]. Indeed, after a decade of work since the first publications (Jansen and Van Durme, 2011; Lee and Glass, 2012; Lee et al., 2015), it is only

in the last couple of years that speech segmentation systems (Bhati et al., 2021; Peng and Harwath, 2022; Kamper, 2023; Algayres et al., 2022b) have successfully done better than a uniform baseline. These progress have enabled the use of discovered spoken words as inputs to spoken language models to learn high-level semantic and syntactic representations(Algayres et al., 2022b) as well as generate intelligible and meaningful spoken sentences (Algayres et al., 2023). The authors highlight that the discovery of boundaries that are aligned with real word boundaries strongly benefits downstream spoken language models.

We present a speech segmentation method inspired by a phoneme segmentation model Strgar and Harwath (2022) that leverages pseudo labelling and recent progress in self-supervised learning (SSL) speech models (Baevski et al., 2020; Hsu et al., 2021; van den Oord et al., 2018; Chen et al., 2021; Babu et al., 2021; Conneau et al., 2020). SSL models are trained on large speech datasets to predict masked parts of the input speech signal. Such pre-training methods yield speech representations that can be quickly adapted through fine-tuning to a variety of downstream tasks ranging from ASR to speaker recognition, keyword spotting, intent classification and emotion recognition (Yang et al., 2021). We exploit the ability of SSL models to learn new tasks quickly and fine-tune a pre-trained XLS-R model (Babu et al., 2021) to predict the word boundaries produced by an off-the-shelf unsupervised speech segmentation system. Our method is inspired by the semi-supervised learning literature (Xie et al., 2019; Yalniz et al., 2019; Scudder, 1965; Hinton et al., 2015; Grill et al., 2020; Chen and He, 2020), that have explored how a model can bootstrap itself by providing its own labels. We applied our method on the three state-of-the-art speech segmentation system: VG-HuBERT, DPDP and DP-Parse (Kamper, 2023; Peng and Harwath, 2022; Algayres et al., 2022b)

---

[1]Code is available at https:// gitlab.cognitive-ml.fr/ralgayres/wav2boundaries

[2]Text segmentation into words is the task of finding word boundaries in a phonemicised text where spaces between words have been removed

and consistently improves their segmentation performances. Our method works particularly well with DP-Parse: on average, over five corpora featuring different languages, an XLS-R fine-tuned on DP-Parse boundaries increases by 130% the segmentation performances compared to the previous state-of-the-art. Finally, our method gets results above the state-of-the-art even in a zero-shot setting where XLS-R is not fine-tuned and sometimes not even pre-trained on the target language.

# 1 Related works

## 1.1 Speech Segmentation

A particularly successful approach that has been applied to the problem of text segmentation is non-parametric Bayesian models (Goldwater et al., 2009; Johnson et al., 2007). This approach has inspired two recent speech segmentation systems: DP-Parse and DPDP (Algayres et al., 2022b; Kamper, 2023). These models segment a spoken sentence by first assigning every speech fragment a probability to be a word. Then, using dynamic programming beam search, one of the most probable segmentation for the whole spoken sentence is sampled. DPDP assigns probability scores using the loss value of an RNN auto-encoder that has been trained to reconstruct random speech sequences. DP-Parse computes probabilities by estimating the frequency of speech fragments using density estimation on speech fragments encoded into Speech Sequence Embeddings (SSE). SSE models are trained with contrastive learning (Algayres et al., 2022a; Settle and Livescu, 2016) or auto-encoders (Kamper, 2018; Peng et al., 2020) to embed variable-size speech segments into fixed-size vectors. DP-Parse authors have shown that using better SSEs (that can be obtained with weak textual supervision) leads to higher segmentation performances.

A second type of model has reached the state-of-the-art in speech segmentation: VG-HuBERT (Peng and Harwath, 2022; Peng et al., 2023). This multimodal model fine-tunes the CLS tokens of a pre-trained HuBERT (Peng and Harwath, 2022) and a pre-trained ViT (Dosovitskiy et al., 2020) on aligned pairs of Engish utterances and images.

Lastly, Fuchs and Hoshen (2023) also tackles speech segmentation into words using pseudo labeling and SSL fine-tuning. Yet, our method finetunes the full XLS-R model using various optimization methods (iterative self-labelling, lr scheduler, data augmentation and loss selection) whereas Fuchs and Hoshen (2023) train a single fully connected layer on top of a frozen Wav2vec2.0 (Baevski et al., 2020) without iterative self-labelling. Also, our method is tested across different languages whereas their model only focuses on English speech.

## 1.2 Wav2vec2.0 and XLS-R

Speech Self-Supervised Learning (SSL) is a paradigm that enables to train deep neural networks directly on the speech stream, typically by predicting masked parts of the input speech signal. Wav2vec2.0 (Baevski et al., 2020) is a particularly performant SSL model that is composed of a convolutional front-end and a stack of transformer layers. Even though other SSL models have outperformed Wav2vec2.0 (Chen et al., 2021; Hsu et al., 2021; Chung et al., 2021) on downstream tasks, Wav2vec2.0 has recently been trained in multilingual settings with XLSR53 ((Conneau et al., 2020), 53 languages), and XLS-R ((Babu et al., 2021), 128 languages). These multilingual SSL models are excellent candidates for our work as we wish to perform speech segmentation into words in different languages. We carry out experiments with XLS-R, one of the latest multilingual Wav2vec2.0 model[3]. We provide in Appendix our experiments with other mono-lingual and multi-lingual Wav2vec2.0 models to analyse the effect of the amount of pre-training data.

# 2 Method

Let us use a speech dataset $C$, a pre-trained speech SSL model $W$, and an off-the-shelf speech segmentation system $S$. On top of $W$ is added a random feed-forward layer with one neuron and a sigmoid activation. Here is our method to train $W$ on boundaries produced by $S$.

First, $S$ is used to infer word boundaries for every spoken sentence in $C$. In addition, spoken sentences are data-augmented with a random quantity of reverb, pitch, time-stretch and time-drop and then encoded into a series of frames by $W$. For each output frame, we create a label that is either 1 if the frame aligns with a word boundary or 0 if not. Because word boundaries are, by nature, not clearly defined in the time domain, we label as 1 the left and right neighbouring frames of every frame that has been already tagged as 1. $W$ is

---

[3]A Wav2Vec2.0 model pre-trained on almost 4000 languages has recently been released but we did not have time to include it in our analysis (Pratap et al., 2023)

fine-tuned with back-propagation by minimizing the negative cross entropy between $W$'s output and the labels. In a sentence, most of the frames are labelled as zeros (for 'not a boundary'), and the loss is particularly low on those frames. To force the model to focus on harder sections of the input, we only backpropagate the loss on the top 50% frames with the highest loss.

At inference, for a given sentence, $W$ produces at each frame the probability of discovering a boundary. To decide which frames should be labelled as a boundary, we apply a peak-detection method that finds local maxima by comparison of neighbouring probability values. This function has two hyperparameters: maximal height of peaks and minimal distance between two peaks. We fit these hyperparameters on the development set by maximizing the F1 scores between the boundaries produced by $S$ and the new boundaries produced by $W$.

At this stage, $W$ can be used to infer on the dataset $C$ a set of new word boundary labels. $W$ is set back to its initial unfine-tuned state and is fine-tuned again on the new word boundaries. This process is iterated until segmentation performances start to decrease.

## 3   Datasets, Evaluation and Hyperparameters

The metric that we use to evaluate performance is the token-F1 score which is the F1 score on correctly discovered tokens. A token is correctly discovered when both its boundaries correspond to the boundaries of a word in the time-aligned transcription. This metric was introduced by the ZeroSpeech Challenge 2017 (Dunbar et al., 2017) and is computed with the TDEv2 library.

The datasets used in this work are the five corpora introduced in the ZeroSpeech Challenge 2017 (REF): Mandarin (2h), French (20h), English (30h), German (19h30min), Wolof (2h43min). The spoken sentences are split into voice activity detections (VAD) (i.e. sequences that only contain speech). The datasets are not split into train/test because they are meant for unsupervised learning. Yet, in order to fit hyper-parameters, the five corpora are split into two sets: Mandarin, French and English are development datasets where hyper-parameters are tuned and German and Wolof are test datasets where the model is tested to show generalization to new languages. We

performed a grid-search on hyper-parameters on the development datasets to maximise the token-F1.

During our fine-tuning step, we use backpropagation on batches that contain 12 spoken sentences of a maximum of 20 seconds each. The XLS-R is fine-tuned on a single 32Go GPU for a maximum of 2000 updates, after which we keep the model with the lowest loss on the development set. Optimization is done with Adam optimizer (Kingma and Ba, 2017), the learning rate is warmed up from 0 to $10^{-4}$ and then decayed back to 0 with a cosine annealing (Loshchilov and Hutter, 2016) with period $10^3$. We freeze the convolutional front end and use 10% dropout, 15% layer-drop, and 15% masked frames. Data augmentation is done mainly with the WavAugment library. We use the values of parameters advised by the authors Kharitonov et al. (2020): for reverb, we sampled the *room scale* uniformly in [0,100] while keeping the other parameters unchanged and for pitch, we pick a value uniformly in the range [-300,300]. Time-stretch coefficients are uniformly sampled between 0.8 and 1.

## 4   Results

### 4.1   F1 scores for different noisy boundaries

In Table 1, we show the comparison of token-F1 scores for different speech segmentation systems and the token-F1 scores after iterative fine-tuning of XLS-R initialised by these systems. Instead of fine-tuning a different XLS-R model on each dataset, we realised that we always get equal or higher performances if we fine-tune only one XLS-R on the boundaries of the five datasets at once[4]. We argue that there is no overfitting possible as our method is completely unsupervised, and no true word boundary labels are used to train those models. In particular, Germand and Wolof datasets have never been used to tweak hyper-parameters. In Appendix 5, we provide the main scores obtained when XLS-R is only fine-tuned on each dataset separately. Even in this setting, our method produces an average token F1 score that is twice higher than the previous state-of-the-art.

The first two segmentation systems are baselines models. For the first, speech is segmented along the VAD timestamps. For the second one, in addition to the VAD timestamps, we added random boundaries

---

[4] we did not use language tags nor language-specific heads.

|  | Mandarin | | French | | English | | German | | Wolof | | average | |
|---|---|---|---|---|---|---|---|---|---|---|---|---|
|  | init | ft | init | ft | init | ft | init | ft | init | ft | init | ft |
| VADs | 4.5 | 2.0 | 4.4 | 1.8 | 4.5 | 2.0 | 3.3 | 1.3 | 1.5 | 1.2 | 3.6 | 1.7 |
| Random | 12.1 | 3.1 | 7.7 | 2.0 | 8.1 | 2.5 | 6.7 | 1.8 | 11.4 | 2.5 | 9.2 | 2.4 |
| VG-Hubert$^\vee$ (Puy22) | 20.0 | 17.0 | 15.0 | 17.7 | 23.2 | 34.0 | 19.9 | 28.8 | 7.0 | 13.5 | 17.0 | 22.2 |
| DPDP$^\dagger$ (Kam22) | 26.3 | 26.0 | 12.2 | 17.4 | 19.5 | 26.6 | 15.2 | 30.2 | 14.8 | 17.3 | 17.6 | 23.5 |
| GradSeg$^\ddagger$ (Fush23) | 12.2 | 27.5 | 18.2 | 31.3 | 19.4 | 30.2 | 12.5 | 21.4 | 11.8 | 24.9 | 14.8 | 27.1 |
| DP-Parse$^\times$ (Alg22) | 16.0 | **32.0** | 15.3 | **41.8** | 21.9 | **42.5** | 13.4 | **49.5** | 17.5 | **37.8** | 16.8 | **40.7** |
| *weak-sup DP-Parse$^\times$* | *28.2* | *31.9* | *30.9* | *52.9* | *31.3* | *55.9* | *34.4* | *61.9* | *39.2* | *40.7* | *32.8* | *48.6* |
| *Gold* | *100.0* | *53.8* | *100.0* | *78.3* | *100.0* | *85.6* | *100.0* | *88.5* | *100.0* | *54.8* | *100.0* | *72.2* |
| *DP-Parse on text* | *50* | *n/a* | *68.1* | *n/a* | *78.5* | *n/a* | *67.4* | *n/a* | *69.1* | *n/a* | *66.6* | *n/a* |

Table 1: Token-F1 obtained by different segmentation systems ('init' in the table) and after iterative fine-tuning of XLS-R on all datasets at once ('ft' in the table). *weak-sup DP-Parse* is a topline that uses weakly-supervised SSEs instead of unsupervised SSEs. *Gold* is a supervised topline where XLS-R is fine-tuned with the true word boundaries. *DP-Parse on text* is obtained by replacing the speech stream by text without spaces between words. ‡:Fuchs and Hoshen (2023), †:Kamper (2023)∨Peng and Harwath (2022)×:Algayres et al. (2022b)

|  | average token-F1 |
|---|---|
| full model | 40.7 |
| without loss selection | 38.5 |
| without data augmentation | 37.7 |
| without tuning the peak detection | 37.4 |

Table 2: Ablation table: average token-F1 score of segmentation over the five corpora. Each row is an ablation compared to the row above itself.

so that token durations have the same mean and standard deviation as the true word tokens. As each VAD give two true word boundaries, we thought that this information could be enough to kick-start our self-training method. The results show that it is not the case, our method leads to a drop in segmentation scores.

Then, we evaluate the three speech segmentation systems that were presented in the review 1. On average, over the five datasets, our method consistently improves their F1 scores with a clear advantage for DP-Parse which gets an average token F1 score of 40.7.

Finally, we present topline models. For *weak-sup DP-Parse*, we segmented speech with the help of weakly-supervised SSEs instead of unsupervised ones (as explained in Section 1). *weaksup DP-Parse* present higher performances than DP-Parse, which shows that our fine-tuning method could yield even better results, providing better SSE models. For *Gold*, we used the true word boundaries, and XLS-R degrades the segmentation performances on average from 100% to 72.2%[5].

The last topline, *DP-Parse on text*[6], gets 66.6% token-F1. The scores on text data show that our fine-tuning strategy has significantly narrowed the performance gap between speech and text. The scores on *Gold* being higher than *DP-Parse on text* show that our strategy has the potential to bridge the gap between speech and text segmentation completely. To complete our analysis, we provide in Appendix Table 4 the Boundary-F1 scores, which are the F1 scores on correctly discovered boundaries instead of correctly discovered tokens.

Overall, the results show great discrepancies between the initial F1 scores of a model and the performances of a fine-tuned XLS-R. We are not yet able to explain precisely why the fine-tuning of XLS-R on DP-Parse works better than on DPDP and VG-HuBERT. The main reason is certainly that we initially chose hyperparameters to maximise performances of XLS-R when it is fine-tuned on DP-Parse. Then, we tried other hyper-parameters to try to boost DPDP and VG-HuBERT scores (different learning rates and data augmentations) but did not manage to improve over the scores reported in Table 1. Figure 2 is a visual presentation of the performances compared to the speech segmentation systems submitted to the Zerospeech challenge (Dunbar et al., 2017) since 2017. The increase in performance obtained by fine-tuning XLS-R appear in deep blue.

To analyse the importance of each of the tricks that we have used in our fine-tuning strategy, we provide in Table 2 the average token-F1 scores over the five datasets by successively ablating each trick. Overall, this table shows that the main gain

---

[5]As the datasets do not have a validation set, when we train XLS-R to predict true word boundaries (and only in this case), we keep a held-out development set to compute the F1 scores.

[6]DP-Parse is applied on the transcriptions of the ZeroSpeech datasets instead of the speech signal

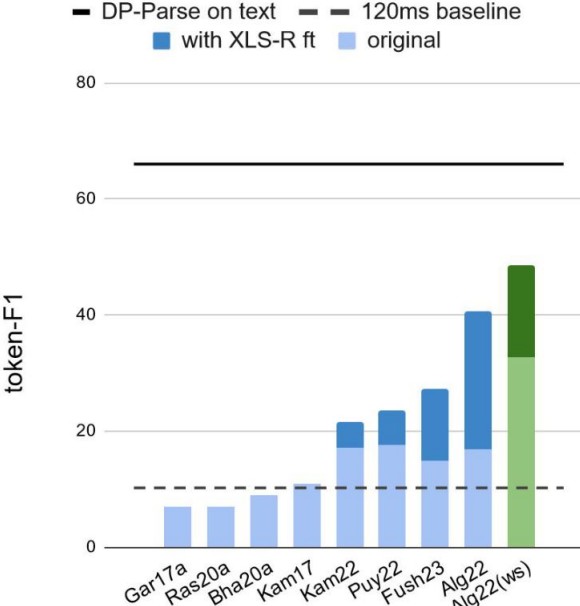

| | Mandarin | French | English | German | Wolof | average |
|---|---|---|---|---|---|---|
| DP-Parse | 35.6 | 38.0 | 22.3 | 46.4 | 23.8 | 33.1 |
| Gold | 24.2 | 60.4 | 62.6 | 66.7 | 22.9 | 53.2 |

Table 3: Zeroshot performances: for each corpus, we report the token-F1 scores after finetuning XLS-R to predict the boundaries of all other corpora except itself. Scores are presented for DP-Parse boundaries and true word boundaries (Gold).

Figure 1: A general view of the performances of speech segmentation models so far. The figure shows the average token-F1 scores (Mandarin, French, English, Wolof, German) obtained by different systems. In light blue are the original scores and in deep blue the increase in performances after XLS-R finetuning. The baseline is a segmentation every 120ms and the topline is DP-Parse applied on text data. In green is the performance obtained by the weakly supervised DP-Parse model.

of our method is obtained by the simple finetuning of XLS-R on noisy boundaries.

### 4.2 Zeroshot performances

We show in Table 3 that our method can be used to segment languages that are unseen during the fine-tuning stage and also unseen during the pre-training stage of XLS-R (which is the case of Wolof). In turn, we selected four out of the five datasets for fine-tuning and used the remaining dataset for testing. We did this experiment using DP-Parse boundaries and the true word boundaries. The results of the zero-shot DP-Parse are sometimes as high as when all datasets are included in the fine-tuning stage. This result echoes the intuition from Peng et al. (2023) that these models can learn universal segmentation features, which appear in their study to coincide with syllables.

### 5 Conclusion

In this work, we propose an unsupervised speech segmentation system that fine-tunes XLS-R on boundaries provided by an external off-the-shelf speech segmentation system. Our method increases word segmentation performances by 130% compared to the previous state-of-the-art. Our method also shows high performances in the zero-shot setting, which suggest that universal segmentation features exist in the speech signal. Regarding interpretation, our results are sometimes hard to explain. Even though we proved that high initial F1 scores (obtained with weak supervision) do lead to better performances, more work is needed on the evaluation of the speech segmentation model to understand what characteristics are beneficial to XLS-R finetuning but that are not captured by F1 scores.

### Limitations

Our method has only been tested on the ZeroSpeech corpora, which come pre-segmented into Voice Activity Detection (VAD). These VADs have been obtained by first force-aligning audio and transcriptions and then by excluding all audio sections that were aligned to silences or noise. If our model is used to segment other audio files, it is important to properly remove beforehand silent sections as well as any other non-speech sections (we advise using Pyannote (Bredin et al., 2019) or Brouhaha (Lavechin et al., 2023) if you cannot rely on force-alignment). Also, the audio from the ZeroSpeech corpora are studio recorded, which means the level of noise is extremely low. The performances of our model with noisier recording conditions would be much lower than those reported in this paper.

### Ethics Statement

Our model inherits from all the biases of audio models pre-trained on a large amount of data. In particular, languages and accentuations that were not present in the original pre-training dataset may be less well encoded by XLS-R, which could result in impaired performances. The reader can refer to the list of pre-training languages in Babu et al. (2021).

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

## A Study on the types of pre-training

Wav2vec2.0 comes in several mono-lingual and multi-lingual pre-trained versions. The monolingual Wav2vec2.0 are *W2V2-LS* trained on LibriSpeech (Panayotov et al., 2015) (960 hours of English speech), and *W2V2-LV* trained on LibriLight (Kahn et al., 2019) (58k hours of English speech). The multilingual versions are *XLSR53* trained on 56k hours of speech from 53 languages, with 82% of the total being English speech and *XLS-R*, trained on 436k hours from 128 languages, with 15% of the total being English speech. Thanks to those pre-training variations we will be able to study the impact of pre-training for the task of speech segmentation into words. All models have the same architecture (convolutions and 24 transformer layers), number of parameters (300M), and training loss (NTXent), they only differ by their pre-training dataset.

Table 6 presents our analysis of the amount of pre-training data required to reach the XLS-R high segmentation performances. We provide the F1 scores after fine-tuning either on DP-Parse or on

| | Mandarin | | French | | English | | German | | Wolof | | average | |
|---|---|---|---|---|---|---|---|---|---|---|---|---|
| | init | ft | init | ft | init | ft | init | ft | init | ft | init | ft |
| VADs | 47.8 | 45.4 | 47.7 | 37.8 | 46.5 | 39.7 | 47.6 | 37.4 | 48.4 | 45.9 | 47.6 | 41.24 |
| Random | 54.3 | 48.1 | 47.0 | 41.0 | 46.3 | 42.2 | 43.3 | 39.5 | 52.9 | 49.4 | 48.9 | 44.0 |
| VG-Hubert[∨] | 63.9 | 63.2 | 55.3 | 57.7 | 59.4 | 65.6 | 58.1 | 64.6 | 51.9 | 54.6 | 57.7 | 61.4 |
| DPDP[†] | 68.3 | 70.2 | 53.5 | 58.6 | 57.5 | 63.7 | 55.6 | 68.5 | 59.6 | 59.2 | 58.9 | 64.1 |
| DP-Parse[×] | 59.9 | **75.5** | 55.9 | **75.4** | 60.0 | **74.1** | 51.5 | **79.8** | 59.0 | **74.1** | 57.3 | **75.8** |
| zero-shot DP-Parse | 59.9 | 62,9 | 55.9 | 71,3 | 60.0 | 69,7 | 51.5 | 75,6 | 59.0 | 72,9 | 57.3 | 70,48 |
| *weak-sup DP-Parse[×]* | *69.9* | *75.4* | *70.0* | *77.6* | *68.4* | *80.8* | *64.4* | *85.2* | *75.5* | *79.4* | *69.6* | *79.7* |
| *Gold* | *100* | *84.4* | *100* | *90.2* | *100* | *93.1* | *100* | *95.2* | *100* | *82.3* | *100* | *89.0* |
| DP-Parse (text) | 76 | n/a | 84,3 | n/a | 89,8 | n/a | 83,5 | n/a | 84,1 | n/a | 83,5 | n/a |

Table 4: Boundary-F1 obtained by different segmentation systems ('init' in the table) and after iterative fine-tuning of XLS-R **on all datasets at once**('ft' in the table). *weak-sup DP-Parse* is a topline that uses weakly-supervised SSEs instead of unsupervised SSEs. Finally, *Gold* is a supervised topline where XLS-R is fine-tuned with the true word boundaries. †:Kamper (2023)∨Peng and Harwath (2022)×:Algayres et al. (2022b)

| | Mandarin | | French | | English | | German | | Wolof | | average | |
|---|---|---|---|---|---|---|---|---|---|---|---|---|
| | init | ft | init | ft | init | ft | init | ft | init | ft | init | ft |
| Token-F1 | | | | | | | | | | | | |
| VG-Hubert[∨] | 20.0 | 15.3 | 15.0 | 16.7 | 23.2 | 34.3 | 19.9 | 27.6 | 7.0 | 1.2 | 17.0 | 19.1 |
| DPDP[†] | 26.3 | 30.2 | 12.2 | 16.1 | 19.5 | 27.0 | 15.2 | 28.9 | 14.8 | 19.1 | 17.6 | 24.2 |
| DP-Parse[×] | 16.0 | **30.4** | 15.3 | **28.8** | 21.9 | **47.3** | 13.4 | **38.2** | 17.5 | **30.2** | 16.8 | **35.1** |
| Boundary-F1 | | | | | | | | | | | | |
| VG-Hubert[∨] | 63.9 | 64.1 | 55.3 | 57.5 | 59.4 | 67.1 | 58.1 | 63.2 | 51.9 | 35.9 | 57.7 | 55.9 |
| DPDP[†] | 68.3 | 73.1 | 53.5 | 57.3 | 57.5 | 63.8 | 55.6 | 68.3 | 59.6 | 59.9 | 58.9 | 62.3 |
| DP-Parse[×] | 59.9 | **73.7** | 55.9 | **68.2** | 60.0 | **76.6** | 51.5 | **73.3** | 59.0 | **70.8** | 57.3 | **72.5** |

Table 5: Token-F1 and Boundary-F1 obtained by different segmentation systems ('init' in the table) and after iterative fine-tuning of XLS-R **on each dataset separately** ('ft' in the table). †:Kamper (2023)∨Peng and Harwath (2022)×:Algayres et al. (2022b)

true boundaries. For visualization of these results, Figure 2 shows the average token-F1 per model. As expected, pre-training Wav2vec2.0 is strongly beneficial for learning word boundaries. Yet, the amount of speech data available for pre-training does not correlate well with segmentation performances. In spite of being trained on much fewer data than W2V2-LV and XLSR53, W2V2-LS reaches high segmentation performances. These Preliminary results on other SSL models have not been included in this section for lack of time. In particular, Hu-BERT (Hsu et al., 2021) has slightly lower performances than Wav2vec2.0 models. Also, a recent Wav2vec2.0 model came to our attention (Pratap et al., 2023), pretrained on nearly 4000 different languages but we did not have time to include this model in our work.

## B   Study on the type of input boundaries

We are not yet able to explain precisely why the fine-tuning of XLS-R on DP-Parse works better than on DPDP and VG-HuBERT. As said in the main paper, the main reason is certainly that the optimisation hyperparameters have been tuned to maximise performances when XLS-R is being fine-tuned on DP-Parse. Yet, we think there could be another reason: the difference in *tokens per type* ratios. This ratio is obtained by first transcribing the discovered speech tokens and then by dividing the number of discovered tokens by the number of different types of transcriptions. Indeed, as XLS-R needs to (at least partially) memorise the different word types that it is trained to segment, XLS-R will more easily learn to segment a small number of types than a large number of types. In Table 7, we show that DP-Parse has a higher *tokens per type* ratio (i.e. fewer types ) than DPDP and VG-HuBERT. For that reason, we think that, compared to its competitors, DP-Parse provides a better kind of input for XLS-R finetuning. This higher *tokens per type* could come from DP-Parse higher *tokens per second*, as shown in Table 7. More work is needed to know if XLS-R simply favours segmentation systems that tend to oversegment (and therefore have higher *tokens per second* and higher *tokens per type*).

For completeness, we also provide in Table 7 the *tokens per second* and *tokens per type* after fine-tuning XLS-R on unsupervised boundaries. As expected from the token-F1 and boundary F1 analysis, fine-tuning XLS-R pushes *token per second*

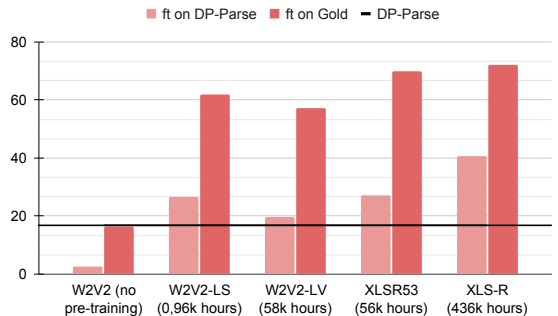

Figure 2:   Average token-F1 scores (Mandarin, French, English, Wolof, German) obtained by different Wav2vec2.0 models pre-trained on different amount of speech and fine-tuned on either DP-Parse boundaries or Gold (i.e. true word boundaries). The average token-F1 score of DP-Parse on the five datasets is represented by a black line.

and *tokens per type* ratios closer to the ratios obtained with true segmentation.

|  | Mandarin | French | English | German | Wolof | *avg* |
|---|---|---|---|---|---|---|
| *W2V2 (no pre-training)* |  |  |  |  |  |  |
| *pre-training hours* | *0* | *0* | *0* | *0* | *0* |  |
| ft on DP-Parse | 1.2 | 0.7 | 7.8 | 1.3 | 1.7 | *2.5* |
| ft on Gold | 27.6 | 6.2 | 12.5 | 13.2 | 22.6 | *16.3* |
| *W2V2-LS* |  |  |  |  |  |  |
| *pre-training hours* | *0* | *0* | *960* | *0* | *0* |  |
| ft on DP-Parse | 21.8 | 24.3 | 44.2 | 18.1 | 24.2 | *26.5* |
| ft on Gold | 57.8 | 52.0 | 85.5 | 63.67 | 51.2 | *62.0* |
| *W2V2-LV* |  |  |  |  |  |  |
| *pre-training hours* | *0* | *0* | *57700* | *0* | *0* |  |
| ft on DP-Parse | 18.5 | 20.2 | 35.2 | 16.8 | 23.4 | *19.7* |
| ft on Gold | 51.1 | 49.5 | 76.0 | 57.2 | 52.3 | *57.2* |
| *XLSR53* |  |  |  |  |  |  |
| *pre-training hours* | *32* | *1424* | *46009* | *1966* | *0* |  |
| ft on DP-Parse | 28.9 | 24.4 | 37.7 | 20.3 | 24.0 | *27.1* |
| ft on Gold | 49.1 | 70.1 | 85.5 | 86.3 | 58.9 | *70.0* |
| *XLS-R* |  |  |  |  |  |  |
| *pre-training hours* | *90* | *23900* | *69500* | *25300* | *0* |  |
| ft on DP-Parse | 32.0 | 41.8 | 42.4 | 49.7 | 37.8 | *40.7* |
| ft on Gold | 53.8 | 78.3 | 85.6 | 88.5 | 54.8 | *72.2* |

Table 6: Token-F1 scores obtained by different Wav2vec2.0 models pre-trained on different amounts of speech and fine-tuned on either DP-Parse boundaries or Gold (i.e. true word boundaries)

| | **Mandarin** | | **French** | | **English** | | **German** | | **Wolof** | | **average** | |
|---|---|---|---|---|---|---|---|---|---|---|---|---|
| | init | ft | init | ft | init | ft | init | ft | init | ft | init | ft |
| *Tokens per type* | | | | | | | | | | | | |
| VG-HuBERT[∨] | 2.08 | 2.43 | 4.40 | 7.38 | 4.03 | 6.56 | 4.05 | 6.62 | 1.51 | 1.08 | 3.21 | 4.81 |
| DPDP[†] | 1.52 | 2.38 | 3.21 | 4.36 | 4.54 | 6.52 | 2.83 | 4.20 | 1.93 | 2.62 | 2.81 | 4.02 |
| DP-Parse[×] | 2.97 | 4.41 | 5.18 | 3.95 | 3.95 | 5.00 | 3.63 | 4.11 | 3.32 | 6.61 | 3.81 | 4.82 |
| *weak-sup DP-Parse[×]* | *2.38* | *5.51* | *6.20* | *6.30* | *6.08* | *6.21* | *5.91* | *6.45* | *5.26* | *4.70* | *5.20* | *5.80* |
| Gold | *2.50* | *2.91* | *14.02* | *7.89* | *17.8* | *8.35* | *8.04* | *6.09* | *14.86* | *5.40* | *11.44* | *6.13* |
| *Token per seconds* | | | | | | | | | | | | |
| VG-HuBERT[∨] | 3.75 | 5.42 | 3.57 | 3.90 | 3.34 | 3.56 | 3.51 | 3.80 | 2.04 | 0.51 | 3.24 | 3.44 |
| DPDP[†] | 2.64 | 2.92 | 3.22 | 3.18 | 3.70 | 3.73 | 3.24 | 3.09 | 2.96 | 3.27 | 3.15 | 3.24 |
| DP-Parse[×] | 4.44 | 4.03 | 3.74 | 2.68 | 3.23 | 2.97 | 3.47 | 2.91 | 3.88 | 4.39 | 3.75 | 3.40 |
| *weak-sup DP-Parse[×]* | *3.38* | *4.10* | *3.29* | *3.30* | *3.31* | *3.01* | *3.37* | *3.09* | *3.92* | *3.69* | *3.50* | *3.40* |
| *Gold* | *2.78* | *3.19* | *3.10* | *3.07* | *3.34* | *3.33* | *2.73* | *2.79* | *3.67* | *3.88* | *3.12* | *3.25* |

Table 7: Token per type and token per seconds obtained by different segmentation systems ('init' in the table) and after iterative fine-tuning of XLS-R **on all datasets at once**('ft' in the table). *weak-sup DP-Parse* is a topline that uses weakly-supervised SSEs instead of unsupervised SSEs. Finally, *Gold* is a supervised topline where XLS-R is fine-tuned with the true word boundaries. †:Kamper (2023)∨Peng and Harwath (2022)×:Algayres et al. (2022b)