# OpenReview forum: "XLS-R fine-tuning on noisy word boundaries for unsupervised speech segmentation into words"
_EMNLP/2023/Conference — EMNLP 2023 Findings_

### Official Review · Reviewer_j6sr · 2023-07-21

**Soundness:** 4

**Excitement:**

4: Strong: This paper deepens the understanding of some phenomenon or lowers the barriers to an existing research direction.

**Missing References:**

Citations to two relevant papers are missing: [1] and [2]. But it's worth mentioning that [2] is published within 3 months of this paper.

Both two papers and this work use pseudo-labels to finetune self-supervised speech models for speech segmentation.

[1] focused on unsupervised phoneme segmentation, by finetuning w2v2 and hubert on pseudo labels generated by off-the-shelf unsup phoneme segmentation models.

[2] showed that using the features of the SSL model itself to produce pseudo label can already bring significant improvement.

[1] L. Strgar and D. Harwath, "Phoneme Segmentation Using Self-Supervised Speech Models," 2022 IEEE Spoken Language Technology Workshop (SLT), Doha, Qatar, 2023, pp. 1067-1073, doi: 10.1109/SLT54892.2023.10022827.

[2] T. S. Fuchs and Y. Hoshen, "Unsupervised Word Segmentation Using Temporal Gradient Pseudo-Labels," ICASSP 2023 - 2023 IEEE International Conference on Acoustics, Speech and Signal Processing (ICASSP), Rhodes Island, Greece, 2023, pp. 1-5, doi: 10.1109/ICASSP49357.2023.10095363.

**Paper Topic And Main Contributions:**

this work finetuned a multilingual self-supervised speech model using labels generated from off-the-shelf unsupervised word-level speech segmentation systems, namely DPDP, VG-HuBERT, and DP-Parse, for improved word segmentation performance.
Significant performance gain is observed for all three types of pseudo-labels. In particular, using pseudo labels from DP-Parse, finetuning XLS-R brings an improvement of 130%.

This work advanced the state-of-the-art of unsupervised word-level speech segmentation by a large margin, which is important for bridging the gap between text-based and speech-based language systems. This work leads to a deeper understanding of multilingual self-supervised speech models


**Questions For The Authors:**

A. I'd like to see some ablation studies on how different tricks affect the performance (augmentation, smoothing, loss selection, peak detection)
B. When using the time stretch augmentation, does the word boundary label also get stretched?
C. How many iterations of self-training are needed before the model performance start to decrease?



**Reasons To Accept:**

1. the fact that simple finetuning on pseudo labels via the general cross-entropy loss can bring such significant gain is very interestingly, which reveals interesting properties of the pretrained self-supervised speech models.

2. this work examined using pseudo labels generated by a range of very different state-of-the-art unsupervised systems, namely DPDP, VG-HuBERT, and DP-Parse, and observed universal improvement. This indicate the robustness of the proposed approach.

3. Training one model on all languages leads to better performance than training language specific models. Although this has been observed in large scale speech models (e.g. OpenAI's Whisper), it's the first time I observe this phenomenon in small scale studies (the total amount of data is below 80 hours.)

4. a few tricks has being proposed to (potentially) make their approach works better, namely augmentation, smoothing, loss selection, peak detection. These tricks are not novel, but the usage is new and they make sense intuitively. These tricks could be valuable for researchers working on speech segmentation


**Reasons To Reject:**

Although significant improvements are shown, more explanations are desired:

1. with regard to the impressive performance zero-shot DP-Parse in Table 1, why is zero-shot working so well? What kind of words are being predicted? Or is there a pattern?;
2. what leads to the discrepancy between using pseudo-labels from DPDP, VG-HuBERT, and DP-Parse



**Reproducibility:**

5: Could easily reproduce the results.

**Reviewer Confidence:**

5: Positive that my evaluation is correct. I read the paper very carefully and I am very familiar with related work.

**Typos Grammar Style And Presentation Improvements:**

line 228:  32Go GPU -> 32GB GPU
line 257: Germand -> German

---

> ### Author Rebuttal · Authors · 2023-08-23
>
> We thank the reviewer very much for their time and effort into reviewing our paper. Here are the responses to your questions and comments.
>
> 1- With regard to the impressive performance zero-shot DP-Parse in Table 1, why is zero-shot working so well? What kind of words are being predicted? Or is there a pattern?;
>
> The zero-shot performance is a difficult to understand effect that was first pointed out by the authors of VG-HuBERT. Our interpretation is that by simply segmenting speech into syllables (which are more universal then words) the model can reach high segmentation scores. Yet this has not been proved yet and further research is needed.
>
> 2- What leads to the discrepancy between using pseudo-labels from DPDP, VG-HuBERT, and DP-Parse
>
> We have studied the difference in F1, precision, recall, number of types of discovered words, differences in word durations,... and still did not find anything that clearly explain the difference. After speaking about this with colleagues, we will certainly dedicate more time to this question in the future.
>
> 3- I'd like to see some ablation studies on how different tricks affect the performance (augmentation, smoothing, loss selection, peak detection) B. When using the time stretch augmentation, does the word boundary label also get stretched? C. How many iterations of self-training are needed before the model performance start to decrease?
>
> We will add an ablation table in the appendix for the camera ready version. Also, yes when applying time stretch we also stretch the labels to keep the match between speech and labels. And finally, the self-training takes usually 3 iterations before starting to degrade. We will include this in the final paper.
>
> 4 - Citations,
> Indeed we forgot to cite [1] and [2] that are very related to our work. Thank you very much for pointing that out.
>
> Thank you very much for all your comments that will make this paper much clearer and much easier to read.

---

### Official Review · Reviewer_gXyg · 2023-07-29

**Soundness:** 4

**Excitement:**

3: Ambivalent: It has merits (e.g., it reports state-of-the-art results, the idea is nice), but there are key weaknesses (e.g., it describes incremental work), and it can significantly benefit from another round of revision. However, I won't object to accepting it if my co-reviewers champion it.

**Paper Topic And Main Contributions:**

This paper proposes to finetune the XLS-R model in an "unsupervised" way to get better performance on word boundary detection. It compares several systems using their methods.

**Questions For The Authors:**

See RR.
Also, I am not convinced why through the described training method, we can get those improvements in token-F1. We did not get any further supervision signal, and there are no new mechanisms introduced. Can you explain why we get that improvement clearly?

**Reasons To Accept:**

Exploring unsupervised ways to perform word segmentation in speech is an interesting direction

**Reasons To Reject:**

1. I wonder if this is unsupervised, as it still receives supervision signals from a trained system. Training that system requires supervised labels, and it is uncertain where we can get those labels if the experiments are unsupervised.
2. This paper does not compare to any force-alignment systems. The task, in my understanding, can be achieved reasonably easily using force alignment with a GMM-based system.
3. Why is the system trained on Gold reference achieved 100.0 token-F1? So is that system already perfect? What on earth is the evaluation metric? How is it computed?

**Reproducibility:**

3: Could reproduce the results with some difficulty. The settings of parameters are underspecified or subjectively determined; the training/evaluation data are not widely available.

**Reviewer Confidence:**

4: Quite sure. I tried to check the important points carefully. It's unlikely, though conceivable, that I missed something that should affect my ratings.

---

> ### Author Rebuttal · Authors · 2023-08-23
>
> We thank the reviewer very much for their time and effort into reviewing our paper. We think there has been a misunderstanding on the objective of this paper. We will make some points clearer in the camera-ready version. Here are the responses to your questions and comments.
>
> 1 - "I wonder if this is unsupervised, as it still receives supervision signals from a trained system. Training that system requires supervised labels, and it is uncertain where we can get those labels if the experiments are unsupervised."
>
> Our model is indeed totally unsupervised, thank you for bringing this up, we will make it clearer in the camera-ready version. As a reminder, the model is a pretrained XLS-R that is fine-tuned on word boundaries produced by an off the shelf speech segmentation system. XLS-R is unsupervised because it is trained with by masking and predicting random part of the input speech signal. The word boundaries are unsupervised labels because they are produced by an unsupervised speech segmentation system (DP-Parse, DPDP, VG-HuBERT). The pipeline is therefore completely unsupervised: no labels obtained with textual supervision were used at any point of the whole process. (Except for the ‘Gold’ topline in Table 1 that serves to show maximal potential performances)
>
> 2 - "This paper does not compare to any force-alignment systems. The task, in my understanding, can be achieved reasonably easily using force alignment with a GMM-based system."
>
> Indeed, force-alignment system would make a very good supervised topline system. Yet, as stated in the introduction, our goal is to mimic infant language acquisition, who are known to learn speech segmentation during their first year of life without explicitly using prior linguistic knowledge. Therefore, we need to avoid using text at any point in the pipeline, which makes it not relevant to us to compare to force-alignment systems that leverage text data.
>
> 3 - “Why is the system trained on Gold reference achieved 100.0 token-F1? So is that system already perfect? What on earth is the evaluation metric? How is it computed?”
>
> I think there is a misunderstanding on what the scores in table 1 are. We will make this clearer in the updated version.  As stated in the description of the table 1, the metric is the token-F1. This metric is indeed not explained enough in section 3. Token-F1 is computed for each corpus and each model. A given model is used to segment a complete corpus into a list of pseudo-word tokens. Then the token-F1 is obtained by comparing all segmented tokens with the word alignment reference. A pseudo-word token is correctly predicted if only both of its boundaries match with the reference.
> Then, in table 1, for each corpus, ‘init’ means the token-F1 score obtained by each off-the-shelf segmentation system, and ‘ft’ is the token-F1 score after fine-tuning XLS-R on the boundary produced by such speech segmentation system.
> Finally, it should be clearer now what the ‘Gold’ scores are. First, ‘init’ is 100% because it is the token-F1 of the true word segmentation, then ‘ft’ is the token-F1 obtained by fine-tuning XLS-R on the gold boundaries (which, as expected, decrease performances from 100% to 72% in average)
>
> 4 - “Also, I am not convinced why through the described training method, we can get those improvements in token-F1. We did not get any further supervision signal, and there are no new mechanisms introduced. Can you explain why we get that improvement clearly?”
>
> This method is common in the weakly-supervised and self-supervised literature. A model can bootstrap itself using a simple pipeline: first produce labels with a given model, then filter some of the labels out, and finally retrain the original model on the labels. This method is the base principle of models like  BYOL https://arxiv.org/abs/2006.07733 . Because of the constraint on the number of pages, we had to remove the related works on weakly supervised learning which, we agree, should be added in the appendix for the camera ready version.
>
> 5- “Reproducibility: 3”
>
> We have created a github that will be made public upon acceptance that ensures full reproducibility of our results.
>
> Lastly, without being too presumptuous, we believe this work is going to be an important paper in the unsupervised speech segmentation literature as the reported gains have never been that high since people have started working on this problem, a decade ago. We are confident that it will attract the attention of the whole field of unsupervised speech segmentation.
>
> Thank you very much for all your comments that will make this paper much clearer and much easier to read.

---

### Official Review · Reviewer_oKq6 · 2023-08-07

**Soundness:** 3

**Excitement:**

2: Mediocre: This paper makes marginal contributions (vs non-contemporaneous work), so I would rather not see it in the conference.

**Paper Topic And Main Contributions:**

The paper proposes an unsupervised speech segmentation system by fine-tuning the XLS-R speech representation model. Through an iterative training strategy, starting with an off-the-shelf speech segmentation system, the finetuned model shows improved results on the speech segmentation downstream task.

**Questions For The Authors:**

A. The proposed method is simple but efficient, however, the generalization of the fine-tuning approach is not convincing, as shown in Table 1 that the performance of DPDP and Hubert based ones are worse than the baseline for Mandarin. The discrepancy between different SSL speech models should be investigated to improve the quality of this work. For instance, the authors could dive deep to compare the precision and recall, the effectiveness of data augmentation, etc.

B. Is the method robust to the accuracy/performance of the initial pseud-label from an off-the-shell speech segmentation system?

**Reasons To Accept:**

The idea of the proposed method is simple yet yields good results. By fine-tuning the added random feed-forward layer, the paper demonstrates that the pre-trained speech SSL system can be effective for the downstream task of speech segmentation. Additionally, the paper utilizes speech from different languages, which provides a valuable exploration into the understanding of SSL speech models on multilingual speech segmentation tasks.

**Reasons To Reject:**

The proposed method does not convincingly demonstrate its generalization and novelty, mostly showing the effectiveness of fine-tuning a certain speech SSL model on speech segmentation tasks. In addition to the main approach, there are many extra tricks used during training and evaluation, such as post-processing, loss sample selection, and data augmentation steps. It would be clearer to add some analysis or ablation study to convince the readers and show the distinct contributions of these different steps.

**Reproducibility:**

4: Could mostly reproduce the results, but there may be some variation because of sample variance or minor variations in their interpretation of the protocol or method.

**Reviewer Confidence:**

4: Quite sure. I tried to check the important points carefully. It's unlikely, though conceivable, that I missed something that should affect my ratings.

---

> ### Author Rebuttal · Authors · 2023-08-23
>
> We thank the reviewer very much for their time and effort into reviewing our paper. Here are the responses to your questions and comments.
>
> 1 - “The generalization of the fine-tuning approach is not convincing, as shown in Table 1 that the performance of DPDP and Hubert based ones are worse than the baseline for Mandarin. The discrepancy between different SSL speech models should be investigated to improve the quality of this work. For instance, the authors could dive deep to compare the precision and recall, the effectiveness of data augmentation, etc. Is the method robust to the accuracy/performance of the initial pseud-label from an off-the-shell speech segmentation system?”
>
> We think there is a misunderstanding on the objective of the paper, which we will make clearer in the camera -ready version. In the unsupervised speech segmentation literature, people are trying to find any method to segment a given corpus without prior knowledge of the given language. The goal being to top the leaderboard on the token-F1 metric. We provide such a method by fine-tuning XLS-R on DP-Parse boundaries, and get very high token-F1 scores. Then, we show generalization of this method to 5 different corpora and even show that this method can generalize to languages unseen during pretraining and fine-tuning (see zero-shot dpparse in table1). Without being too presumptuous, the scores obtained in our paper are extremely high, and we honestly believe this work is going to be an important paper in the unsupervised speech segmentation literature. The reported gains have never been that high since people have started working on speech segmentation, a decade ago.
>
> Then, for completeness, we also provide the scores of our method applied to DPDP and VGHubert and indeed these latter do not show consistent improvements. Yet, we do not believe that this observation shows that our method does not generalize. For us this shows that the word boundaries produced by DPDP and VG-HuBERT are not good enough to kickstart the self-training procedure. Finding why DP-Parse boundaries help fine-tuning and not DPDP and VG-HuBERT  is the subject of a whole new paper that would compare precision, recall, number of implicit types, average length and duration of words, … this project is still ongoing research in our lab. We will add this clarification in the final paper.
>
> 2 - “ In addition to the main approach, there are many extra tricks used during training and evaluation, such as post-processing, loss sample selection, and data augmentation steps. It would be clearer to add some analysis or ablation study to convince the readers and show the distinct contributions of these different steps.”
>
> Each one of these tricks provide a little bit of improvements. We agree that an ablation table would be appreciated, and we will add one in the appendix for the camera ready version.
>
> Thank you very much for all your comments that will make this paper much clearer and much easier to read.

---

### Meta-Review · Area_Chair_JndP · 2023-09-19

**Recommendation:** 4

**Metareview:**

This paper presents a method for improving unsupervised speech segmentation by leveraging the large pre-trained self-supervised model XLS-R and off-the-shelf unsupervised word segmentation models to produce labels for self-training. The paper studies three existing approaches, namely DP-PARSE, VG-HUBERT and DPDP, and finds that pseudo labels obtained with DP-PARSE can be successfully used to fine-tune the XLS-R model for unsupervised word segmentation, resulting in an improvement of 130%. The authors report discrepancies between different off-the-shelf word segmentation methods and leave the analysis for future work.

The reviewers liked that this simple idea improves results by a large margin and that the paper evaluates the method in multilingual settings. The reviewers asked for several ablations experiments and clarification of some results. The authors ran the requested ablation experiments and tried to clarify the results, noting that some observations will probably need a more thorough analysis in future papers. After reading the paper and reviews, I base my decision on the review by the reviewer j6sr, who works on unsupervised word segmentation and is excited about this paper.

---

### Decision · Program_Chairs · 2023-10-07

**Decision:**

Accept-Findings

**Comment:**

This paper presents a method for improving unsupervised speech segmentation by leveraging the large pre-trained self-supervised model XLS-R and off-the-shelf unsupervised word segmentation models to produce labels for self-training. The paper studies three existing approaches, namely DP-PARSE, VG-HUBERT and DPDP, and finds that pseudo labels obtained with DP-PARSE can be successfully used to fine-tune the XLS-R model for unsupervised word segmentation, resulting in an improvement of 130%. The authors report discrepancies between different off-the-shelf word segmentation methods and leave the analysis for future work.

The reviewers liked that this simple idea improves results by a large margin and that the paper evaluates the method in multilingual settings. The reviewers asked for several ablations experiments and clarification of some results. The authors ran the requested ablation experiments and tried to clarify the results, noting that some observations will probably need a more thorough analysis in future papers. After reading the paper and reviews, I base my decision on the review by the reviewer j6sr, who works on unsupervised word segmentation and is excited about this paper.